# The Seroprevalence of Influenza A Virus Infections in Polish Cats During a Feline H5N1 Influenza Outbreak in 2023

**DOI:** 10.3390/v17060855

**Published:** 2025-06-16

**Authors:** Anna Golke, Tomasz Dzieciątkowski, Olga Szaluś-Jordanow, Michał Czopowicz, Lucjan Witkowski, Monika Żychska, Ewa Domańska, Dawid Jańczak, Tomasz Nalbert, Stephanie Lesceu, Marzena Paszkowska, Justyna Giergielewicz, Tadeusz Frymus

**Affiliations:** 1Department of Preclinical Sciences, Institute of Veterinary Medicine, Warsaw University of Life Sciences-SGGW, Ciszewskiego 8, 02-786 Warsaw, Poland; 2Chair and Department of Medical Microbiology, Medical University of Warsaw, Chałubińskiego 5, 02-004 Warsaw, Poland; dzieciatkowski@wp.pl; 3Department of Small Animal Diseases with Clinic, Institute of Veterinary Medicine, Warsaw University of Life Sciences-SGGW, Nowoursynowska 159c, 02-776 Warsaw, Poland; olga_szalus-jordanow@sggw.edu.pl (O.S.-J.); tadeusz_frymus@sggw.edu.pl (T.F.); 4Division of Veterinary Epidemiology and Economics, Institute of Veterinary Medicine, Warsaw University of Life Sciences-SGGW, Nowoursynowska 159c, 02-776 Warsaw, Poland; michal_czopowicz@sggw.edu.pl (M.C.); lucjan_witkowski@sggw.edu.pl (L.W.); monika_zychska@sggw.edu.pl (M.Ż.); tomasz_nalbert@sggw.edu.pl (T.N.); 5Department of Food Hygiene and Public Health Protection, Faculty of Veterinary Medicine, University of Life Sciences-SGGW, Nowoursynowska 159, 02-776 Warsaw, Poland; ewa_domanska@sggw.edu.pl; 6Department of Infectious and Invasive Diseases and Veterinary Administration, Institute of Veterinary Medicine, Faculty of Biological and Veterinary Sciences, Nicolaus Copernicus University, 87-100 Toruń, Poland; djanczak@umk.pl; 7Innovative Diagnostics, Avian Department, 34790 Grabels, France; stephanie.lesceu@innovative-diagnostics.com; 8Vetlab sp. z o.o., Wodzisławska 6, 52-017 Wrocław, Poland; marzena.paszkowska@vetlab.pl (M.P.);

**Keywords:** highly pathogenic avian influenza (HPAI) H5N1, influenza A virus (IAV), domestic cats, seroprevalence, viral reassortment

## Abstract

Recently, cats have emerged as potential incidental hosts for avian and human influenza A viruses (IAVs), including the highly pathogenic avian influenza (HPAI) H5N1 virus. Following an unprecedented outbreak of H5N1 HPAI in cats in Poland in June 2023, we conducted a cross-sectional epidemiological study to assess the seroprevalence of IAV, especially H5Nx, infections in domestic cats. Eight hundred thirty-five serum samples collected in June 2023 were tested using a competitive ELISA for antibodies to IAV nucleoprotein. Positive or doubtful samples were further screened for H5-specific antibodies. The overall seropositivity for IAV was 8.5% (CI 95%: 6.8%, 10.6%; 71/835 cats), and 23/68 IAV-seropositive cats (33.8%) were also seropositive for H5 antigen. Multivariable analysis identified young age (≤8 years) and male sex as significant risk factors for H5 seropositivity, while non-H5-IAV seropositivity was more common in cats aged ≥12 years. These findings suggest different exposure pathways and host risk profiles for H5 and non-H5 IAVs and underscore the importance of enhanced surveillance in cats, particularly in regions affected by HPAI outbreaks. Given the susceptibility of cats to both avian and human IAVs, including subclinical infections, there is a theoretical risk for viral reassortment. Preventive measures, including vaccinating humans and restricting outdoor access for cats, should be considered in endemic areas.

## 1. Introduction

Domestic cats have been largely undervalued for many years as potential reservoir hosts for influenza A viruses (IAVs). However, the recent global expansion of the highly pathogenic avian influenza (HPAI) H5N1 virus, descended from strains that initially emerged in Southeast Asia, has challenged this assumption [1,2,3,4].

The current epizootic, driven by the clade 2.3.4.4b H5N1 virus, which is believed to have begun in Europe or Central Asia around 2020, has been characterized by rapid intercontinental spread, reaching the Americas and even Antarctica for the first time. It continues to evolve through multiple reassortments and has frequently spilled over into terrestrial and marine mammals [5]. Notably, sustained mammal-to-mammal transmission has been documented in several settings, including mink farms in Europe [6,7], sea lions in South America [8,9], and, most recently and surprisingly, dairy cattle in the United States [10].

Most H5N1 virus infections in domestic cats have so far been sporadic. One such case was reported in France in December 2022, where a domestic cat died following infection with this virus. The cat lived with a human family near a commercial duck farm, which veterinary authorities had classified as possibly HPAI-infected after observing a 20% drop in egg production, as in ducks, this virus often induces subclinical infections. Genomic sequencing of the virus isolated from the cat revealed key mammalian-adaptive mutations, including E627K in the polymerase basic protein 2 (PB2) and E26G in the nonstructural protein 2 (NS2) [11].

This pattern of isolated cases began to shift following the emergence of more significant outbreaks in Poland, South Korea, and, recently, the United States, where infections in farm cats were directly linked to contact with infected dairy cattle and the consumption of raw milk [2,3,4,10,12]. Moreover, a very recent report from the United States has also indicated the transmission of the virus from humans to their pet cats. In May 2024, HPAI H5N1 clade 2.3.4.4b genotype B3.13 infection was detected in two exclusively indoor cats with respiratory and neurologic illness in separate Michigan households. Both owners had occupational exposure to dairy farms with confirmed HPAI H5N1-positive cattle but unfortunately declined influenza A(H5) testing [13].

Clusters of infections in a significant number of cats had been observed much earlier. The epizootic in Poland in 2023 was unprecedented in scale, with HPAI H5N1 virus infection confirmed in 29 domestic cats, one captive caracal, a few pet ferrets, and a dog [3,4,14,15]. However, the number of affected animals was likely higher, with cases reported nationwide. The disease in clinically ill cats was usually severe and rapidly progressing, taking 1–7 days from the onset of clinical signs to death. Genetic analyses of HPAI H5N1 viruses isolated from cats showed nearly identical genomes, suggesting a monophyletic origin and a likely common source of infection [3,4]. All sequences carried mammalian-adaptive mutations in the polymerase gene, specifically PB2-E627K and PB2-K526R, associated with enhanced replication at lower temperatures [2,3,4,14,15].

In South Korea, H5N1 virus feline infections were confirmed at two separate shelters in Seoul during June and July 2023 without any associated human cases. In one outbreak, the source of infection was traced to contaminated poultry-based pet food, while in the other, the origin remained unknown. The latter outbreak was particularly severe, resulting in the death of 38 out of 40 infected cats. Genomic sequencing confirmed infection with clade 2.3.4.4b H5N1 virus carrying the PB2-D701N mutation, a substitution strongly associated with adaptation to mammalian hosts and increased pathogenicity [12].

Although H5N1 virus infections in felids frequently result in death, asymptomatic H5Nx infections in carnivores, including domestic cats, have also been documented. This was observed both before and after the clade 2.3.4.4b of the H5N1 virus emerged. A notable example occurred in 2006 in an Austrian animal shelter, where exposure between an infected swan and resident cats led to the detection of virus-specific RNA in pharyngeal swabs from 3 out of 40 randomly tested cats. None of these cats exhibited clinical symptoms of influenza, and seroconversion was observed in only two individuals [16]. Similarly, a serological survey of feral cats in Florida between 2008 and 2010 detected antibodies against influenza A virus in just 4 of 927 serum samples (0.43%) [17]. These findings suggest that before 2020–2021, the seroprevalence of anti-H5 antibodies in cats and dogs was generally low.

Following the emergence of clade 2.3.4.4b, the first documented case of seroconversion in companion animals occurred during an outbreak of HPAI H5N1 on a backyard poultry farm in Italy in April 2023. Five dogs and one cat residing on these premises developed antibodies despite the absence of clinical signs. The implicated viral strain carried the mammalian-adaptive T271A mutation in the PB2 gene. No transmission to humans was detected following virological and serological monitoring [18].

The seroprevalence of H5N1 IAVs has also been evaluated in dogs with exposure to infected wild birds. In a study conducted in Washington State, USA, from March to June 2023, antibodies to this IAV subtype were detected in 4 out of 194 (2.1%) dogs involved in hunting or training activities with waterfowl. While this supports the potential for transmission from birds to dogs, the low seroprevalence, lack of disease in seropositive animals, and absence of dog-to-dog transmission suggest that clade 2.3.4.4b H5N1 viruses circulating in North America during 2022–2023 were poorly adapted to canine hosts [19].

In Japan, serological surveillance of wild raccoons revealed that 5 out of 114 individuals (4.4%) sampled during 2022–2023 were seropositive for H5Nx, while samples collected in 2019–2021 showed much lower seroprevalence rates (0.3–0.9%). This temporal shift underscores the increased affinity of clade 2.3.4.4b viruses for mammalian hosts [20].

Given the increasing number of H5N1 virus spillover events and the potential for adaptation to mammals, including humans, and considering the often close contact between cats and their owners, it is crucial to investigate the role of cats in the ecology of H5N1 viruses. Therefore, the aim of this seroepidemiological investigation was to determine the seroprevalence of IAV infection in domestic cats during the 2023 outbreak of feline HPAI H5N1 influenza in Poland. As mentioned, the number of confirmed feline clinical cases was 29, but the range of subclinical, mild, or unrecognized infections remains unknown.

## 2. Materials and Methods

### 2.1. Study Design

This descriptive cross-sectional study was based on convenience sampling [21]. Blood samples were collected by veterinarians for routine clinical chemistry analysis and sent to a commercial laboratory (Vetlab sp. z o.o., Wrocław, Poland). Three criteria had to be fulfilled to include a cat in the study: (i) blood collected in June 2023; (ii) basic demographic data available, including sex, breed, and age; (iii) age of at least 6 months to exclude maternal antibodies.

The minimum sample size (the minimum number of animals required to conduct the study) was calculated so that it ensured the estimation of the seroprevalence of IAV infection, assuming an expected true seroprevalence of 50%, precision of estimation of ±5%, level of confidence of 95%, and ELISA diagnostic sensitivity (dSe) of at least 80% and diagnostic specificity (dSp) of at least 90% [21]. The minimum sample size was 777 cats.

All sera were screened for IAV antibodies. Those classified as seropositive were further tested for H5Nx-IAV antibodies.

The seroprevalence study was conducted using anonymized residual blood samples obtained during routine diagnostic procedures. In accordance with applicable institutional and national regulations, the use of such surplus clinical material for research purposes is exempt from the requirement for informed consent from animal owners.

### 2.2. Study Population

The study population comprised 835 cats, 408 males (48.9%) and 427 females (51.1%), aged from 6 months to 21.0 years, with a median (IQR) of 8.0 (4.0–13.0) years. Age did not differ significantly between males and females (*p* = 0.183). Neuter status was known for 188/835 cats (22.5%): 135/188 cats (71.8%) were castrated—significantly more males (100/101; 99.0%) than females (35/87; 40.2%) (*p* < 0.001).

Most of the tested animals (667/835; 79.9%) were domestic short-/longhair (DSH/DLH) cats, while the remaining 168 cats (20.1%) belonged to 24 breeds, of which the most common were British short-/longhair (39 cats; 4.7%), Maine coon (26 cats, 3.1%), Siberian (13 cats, 1.6%), Persian (12 cats, 1.4%), Ragdoll (11 cats, 1.3%), and Russian blue (10 cats, 1.2%). The remaining 18 breeds were represented by less than 10 individuals. Data regarding the diet, type of location, and outdoor access were unavailable.

### 2.3. Serological Tests

Antibodies to IAV were detected using the commercial competitive ELISA for the detection of antibodies against the nucleoprotein of IAV (ID Screen^®^ Influenza A Antibody Competition Multi-Species, IDvet Innovative Diagnostics, Grabels, France), henceforth referred to as the IAV-ELISA. Positive or doubtful samples were then tested using a commercial competitive ELISA for the detection of antibodies against the H5 hemagglutinin of IAV (ID Screen^®^ Influenza H5 Antibody Competition 3.0 Multi-species), henceforth referred to as the H5-ELISA. Both ELISAs were performed according to the manufacturer’s manual. The optical density (OD) was read at a 450 nm wavelength in a microplate reader (Epoch Microplate Spectrophotometer, Agilent, Santa Clara, CA, USA). Based on the competition percentage (ratio of sample OD to negative-control OD; S/N%), samples were classified as follows: in the IAV-ELISA—positive if S/N% ≤ 45%, doubtful if S/N% from >45% to <50%, and negative if S/N% ≥ 50%; in the H5-ELISA—positive if S/N% ≤ 50%, doubtful if S/N% from >50% to <60%, and negative if S/N% ≥ 60%. Cats in this study were considered seropositive if they obtained positive or doubtful results.

### 2.4. True Seroprevalence Estimation

The apparent seroprevalence was defined as the proportion of cats seropositive in the ELISA. Data on the diagnostic accuracy of the ELISAs in cats were not available. The accuracy of the IAV-ELISA has so far been assessed in dogs [22], horses [23], and pigs [24]—the results obtained in these studies are presented in Table 1.

The true seroprevalence was estimated using the Bayesian approach with the Gibbs sampler [25] in EpiTools (100,000 iterations simulated with 10,000 discarded to allow the convergence of the model) [26] and is reported as the median and 95% credible interval (CrI). The prior probabilities for the true seroprevalence were set at α = 1 and β = 1 (non-informed priors) due to the unknown epidemiological situation of the cat population in Poland. Since the diagnostic accuracy of the IAV-ELISA in cats is unknown and the estimates available for dogs [22], horses [23], and pigs [24] substantially differ, the α and β parameters of the prior probability distribution of dSe and dSp were calculated for three different scenarios, each assuming different estimates of the diagnostic accuracy of the IAV-ELISA from the ones correct for cats. Scenario 1 was based on the estimates obtained in dogs, scenario 2 on those in horses, and scenario 3 on those in pigs. The α and β parameters were set at values calculated from the beta distribution, assuming the 95% confidence (or credible) limits from the abovementioned studies as the 2.5th and 97.5th percentiles of the beta distribution.

No data on the diagnostic accuracy of the H5-ELISA were available, so only the seropositivity was reported.

### 2.5. Statistical Methods

As age was not normally distributed (Shapiro–Wilk test: *p* < 0.001), it was summarized using the median, interquartile range (IQR), and range and compared between groups using the Mann–Whitney U test. A receiver operating characteristic (ROC) curve analysis with maximization of Youden’s index was used to identify the most optimal cut-off value for age categorization. Categorical variables were expressed as counts and proportions and compared between groups using Pearson’s χ^2^ test or Fisher’s exact test (if the expected count in any contingency table cell was <5). The 95% confidence intervals (CI 95%) for proportions were calculated using the Wilson score method [27]. The odds ratio (OR) was used to express the strength of association between categorical variables, and the Haldane–Anscombe correction (adding 0.5 to each contingency table cell) was used in the case of zero cells. A multivariable logistic regression with backward stepwise elimination was used to investigate the relationship between serological status and cats’ individual characteristics, which proved significant in the univariable comparisons [28]. The significance level (α) was set at 0.05. All statistical tests were two-tailed. The statistical analysis was performed in TIBCO Statistica 13.3 (TIBCO Software Inc., Palo Alto, CA, USA).

## 3. Results

### 3.1. Prevalence of IAV Antibodies

Of the 835 cats tested with the IAV-ELISA, 68 cats (8.1%) were positive and 3 cats (0.4%) were doubtful, so in total, 71/835 cats were considered seropositive for IAV. The apparent seroprevalence of IAV infection was 8.5% (CI 95%: 6.8%, 10.6%). The estimated true seroprevalence of IAV infection differed between the three scenarios: 7.9% (95% CrI: 5.9%, 10.4%) in scenario 1, 4.0% (95% CrI: 1.0%, 6.9%) in scenario 2, and 3.1% (95% CrI: 0.1%, 9.4%) in scenario 3. However, in none of scenarios was it higher than the apparent seroprevalence.

### 3.2. Prevalence of IAV-H5Nx Antibodies

As 3 serum samples of cats seropositive for IAV were lost, 68 IAV-seropositive cats were eventually tested using the H5-ELISA: 16 cats (23.5%) turned out to be positive, 7 (10.3%) were doubtful, and the remaining 45 (66.2%) were negative. Also in this test, cats with doubtful results were considered seropositive. As a result, depending on the status of the 3 untested cats whose serum samples were lost, the apparent seroprevalence of IAV-H5Nx infection could range from 2.8% (CI 95%: 1.8%, 4.1%; 23/835 cats if 3 lost sera were negative) to 3.1% (CI 95%: 2.1%, 4.5%; 26/835 cats if 3 lost sera were positive).

### 3.3. Association Between Individual Characteristics of Cats and Seropositivity for IAV and IAV-H5Nx

None of the analyzed characteristics of cats were significantly associated with IAV-seropositive status in the univariable analysis, although seroprevalence was almost significantly higher among DSH/DLH (Table 2). Age did not differ considerably between IAV-seropositive (median: 8.0 years, IQR: 5.0–13.0 years) and IAV-seronegative cats (median: 8.5 years, IQR: 4.0–13.0 years; *p* = 0.643).

Two individual characteristics of cats turned out to be significantly associated with IAV-H5Nx-seropositive status in the univariable analysis: a substantially higher proportion of cats seropositive for IAV-H5Nx were males (*p* = 0.015), and cats seropositive for IAV-H5Nx were significantly younger (median: 6.0 years, IQR: 2.0–8.0 years) than H5-seronegative cats (median: 9.0 years, IQR: 4.0–13.0 years; *p* = 0.024). The most optimal threshold age was 8 years—cats aged 8 years or less were significantly more likely to have antibodies to IAV-H5Nx (*p* = 0.002) (Table 3). Both male gender and age ≤ 8 years proved to be significantly and independently positively associated with the seropositive status for IAV-H5Nx infection in the multivariable analysis (Table 4). The highest seroprevalence of IAV-H5Nx infection was observed in ≤8-year-old male cats, and no significant difference in the seroprevalence of IAV-H5Nx infection was found between males > 8 years and females (*p* = 0.230) (Figure 1).

Interestingly, 45 cats seropositive for IAV but H5-seronegative were significantly older (median: 12.0 years, IQR: 6.0–13.0 years) than IAV-H5Nx-seropositive cats (*p* = 0.001) as well as IAV-seronegative cats (*p* = 0.024) (Figure 2).

## 4. Discussion

In our study, the seroprevalence of IAV infection in domestic cats was considerably below 10%, with anti-H5 antibodies detected in approximately 3% of the population. This suggests that most IAV infections in cats were due to subtypes other than H5. From a demographic perspective, younger male cats, particularly those under 8 years old, were at the highest risk for H5Nx infection. Interestingly, H5-seropositive cats were significantly younger than those seropositive for other IAV subtypes. Conversely, IAV infections with non-H5 subtypes were most prevalent in older cats, regardless of sex.

One plausible explanation for the increased seropositivity of H5Nx in younger males is their behavioral tendency for activities such as hunting or territorial exploration, which increases the likelihood of contact with infected wild birds. In Poland, where a substantial proportion of domestic cats (including those in urban areas) have outdoor access, such exposure is likely. Many cat owners resist recommendations for the indoor-only confinement of their pets, which further contributes to this risk [29].

However, behavioral differences alone may not fully explain the observed sex-based disparity. Another possible contributing factor is that male cats may be more frequently engaged in territorial combat with other males, including free-ranging or feral cats that depend on hunting for sustenance and may be exposed to IAVs. Although this hypothesis is speculative, it raises important questions regarding the potential for cat-to-cat transmission and the minimal infectious dose required for seroconversion. To address these questions, future studies should include feral and free-roaming cats, as all animals tested in our study were owned.

The higher seroprevalence of non-H5 IAV infections in older cats may reflect zooanthroponotic transmission from owners with infections. Behavioral studies show that aging cats seek closer physical proximity to their owners. A survey carried out in the United Kingdom involving over 2000 cats aged 11 and older identified increased affection and physical closeness as the most commonly reported age-related behavioral changes [30]. This aligns with previous case reports of symptomatic H1N1 infections in senior cats following close contact with humans with infections [31,32]. However, it is important to note that in our study, we neither determined the specific subtypes of non-H5 IAVs to which cats had been exposed nor investigated whether these viruses were of human origin or pathogenic to humans. Thus, the proposed link between age-related behavior and zoonotic transmission should be interpreted with caution.

To place our findings in a broader European context, it is helpful to compare them with recent data from the Netherlands, where serum samples were collected from both stray and owned cats between 2020 and 2023 [33]. Among owned animals sampled between November 2020 and March 2023, the seroprevalence for H5 was 0.46%, and seroprevalence for H1 was 4.6%. In stray cats sampled between October 2020 and June 2023, H5 seroprevalence reached 11.8%, and H1 seroprevalence was 5%. Interestingly, two cats had antibodies to both H5 and H1, raising concerns about the possibility of simultaneous infection with both subtypes and potential reassortment [33]. H5N1 exposure in cats was also investigated in France by screening 728 outdoor cats (both owned and stray) between December 2023 and January 2025. Antibodies were detected in 19/728 cats (2.6%), with an estimated true seroprevalence of 1.8% (CI 95%: 0.7%, 3.4%). The absence of hunting behavior was identified as the only significant protective factor, which is consistent with our results, confirming that young hunting cats are at greater risk of H5N1 infection [34]. We also suspect that the higher seroprevalence of anti-H5 antibodies in Polish domestic cats than in the Dutch study may be related to the specific period in which the samples were collected (feline H5N1 IAV outbreak in 2023) when cats in Poland could have been exposed to an unidentified source of the virus.

The authors of the Dutch study also suggested that variability in clinical outcomes, from a fatal disease to asymptomatic infection, could be primarily influenced by the infection route and the infectious dose, as was observed in the experimental challenge of cats with a low dose of H5N1 [33,35]. While we strongly agree, we also propose that variant-specific differences within the H5N1 subtype may play an important role, especially the presence of mutations associated with adaptation to mammalian hosts.

A potentially novel perspective on the occurrence of influenza in domestic cats involves considering how human vaccination patterns might indirectly influence feline seroprevalence, particularly regarding non-H5 IAV subtypes. For instance, influenza vaccination coverage in the Netherlands between 2019 and 2023 exceeded 50% among people over 60, compared to approximately 20% in Poland [36,37]. In the 2022/2023 season, the overall influenza vaccination coverage in Poland was only 5.7% [38]. This suggests that Polish domestic cats could have more frequent contact with humans shedding seasonal IAVs than their Dutch counterparts.

This could represent an interesting new angle from which to examine the broader implications of human vaccination on interspecies transmission dynamics. Additionally, keeping cats indoors would minimize their contact with wildlife and lower their risk of acquiring or amplifying influenza viruses.

Given that some subclinical IAV infections in cats can go undetected and that cats are susceptible to both H5N1 and human-derived H1N1 viruses, there is a theoretical risk of viral reassortment within feline hosts. Such reassortment could generate novel strains of influenza virus with enhanced virulence or transmissibility in humans.

The observed anti-H5 seroprevalence in Polish domestic cats raises essential questions about the likely sources of exposure, which we cannot definitively answer. It is possible that outdoor cats encounter infected birds or other wild animals. Since well-fed domestic cats often leave their prey uneaten, owners may remain unaware of such interactions [29].

Our findings suggest a need to reconsider the role of companion animals as potential intermediate hosts in the ecology and evolution of IAVs. The U.S. CDC’s Influenza Risk Assessment Tool (IRAT) and the WHO’s Tool for Influenza Pandemic Risk Assessment (TIPRA) estimate a low pandemic risk for H5N1 clade 2.3.4.4b [39]. However, these tools assess only the current risk and do not predict the virus’s future evolutionary trajectory or reassortment potential. Considering that on March 24, the UK Department for Environment, Food & Rural Affairs and the Animal and Plant Health Agency issued a statement confirming that the Chief Veterinary Officer had identified a case of avian-origin influenza (H5N1) in a sheep in Yorkshire, it is likely that we will soon have to contend with new routes of virus transmission in Europe [40].

We now have more information about the global distribution, host range, and genetic diversity of the H5N1 virus than for most other zoonotic pathogens. So far, most surveillance systems remain biased toward symptomatic or dead animals. In contrast, the COVID-19 pandemic has underscored that subclinical infections play a substantial role in pathogen transmission and contribute to maintaining epidemics at the population level [41].

In light of the above, future research should prioritize the following: (i) including feral and free-roaming cat populations in serological studies; (ii) confirming ELISA results with other methods to determine subtype-specific exposure; and (iii) investigating behavioral risk factors and their relation to infection dynamics through case–control or cohort designs. These steps will help address current methodological limitations and enhance our understanding of IAV ecology in companion animals.

## Figures and Tables

**Figure 1 viruses-17-00855-f001:**
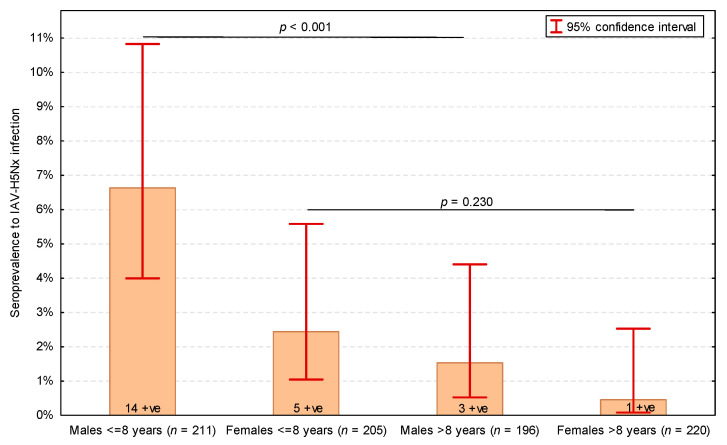
Seroprevalence of influenza A virus subtype H5Nx (IAV-H5Nx) infection in cats depending on sex (males and females) and age (dichotomized into ≤8 years and >8 years). Numbers of seropositive (+ve) cats presented on bars. *p*-values of Pearson’s χ^2^ test presented. The total number of cats was 832, as the serum samples of 3 IAV-seropositive cats were not tested in the H5-ELISA.

**Figure 2 viruses-17-00855-f002:**
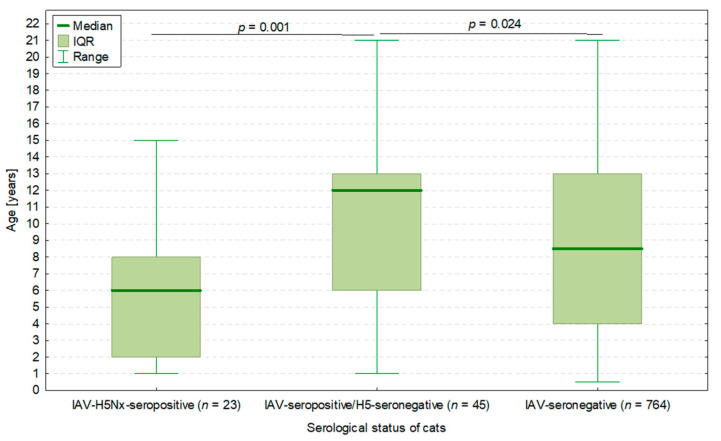
The age of cats in three groups depending on IAV infection status: 23 cats seropositive for both IAV and H5Nx subtype, 45 cats seropositive for IAV but negative for H5Nx subtype, and 764 cats negative for IAV (hence not tested for H5Nx). *p*-values of the Mann–Whitney U test presented. The total number of cats was 832, as the serum samples of 3 IAV-seropositive cats were not tested in the H5-ELISA.

**Table 1 viruses-17-00855-t001:** Prior parameters of the beta distribution of diagnostic sensitivity (dSe) and diagnostic specificity (dSp) of the ELISA detecting antibodies to the influenza A virus (IAV) used for estimating the true seroprevalence of IAV infection in cats in Poland in June 2023.

Scenario	Diagnostic Accuracy of the IAV-ELISA Based on the Literature		Prior Parameters of the Beta Distribution for:		Reference
	dSe (CI 95%) [%]	dSp (CI 95%) [%]	dSe	dSp	
1	94.0 (81.8, 99.1)	98.7 (98.4, 99.0)	α = 27.133 β = 2.058	α = 4936 β = 66	De Benedictis et al., 2010 [22]
2	99.0 (94.5, 99.9)	95.3 (92.7, 97.3)	α = 99.01 β = 1.99	α = 348.85 β = 18.15	Kittelberger et al., 2011 [23]
3	69 (58, 79)	89 (75, 97)	α = 54.82 β = 25.18	α = 34.82 β = 5.18	Tse et al., 2012 [24]

**Table 2 viruses-17-00855-t002:** Univariable analysis of association between individual characteristics of cats and seropositive status for influenza A virus (IAV) infection.

Characteristics	Category	Number (%) of IAV-Seropositive Cats in Category	*p*-Value	Odds Ratio (95% Confidence Interval)
Sex	Females Males	33/427 (7.7) 38/408 (9.3)	0.412	Reference category 1.23 (0.75, 2.00)
Castration (N = 188)	Intact Castrated	3/53 (5.7) 20/135 (14.8)	0.085	Reference category 2.90 (0.82, 10.2)
Breed	Pedigree DSH/DLH ^a^	8/168 (4.8) 63/667 (9.5)	0.052	Reference category 2.09 (0.98, 4.44)

^a^ Domestic shorthair/domestic longhair cats.

**Table 3 viruses-17-00855-t003:** Univariable analysis of association between individual characteristics of cats and seropositive status for infection with influenza A virus of H5Nx subtype (IAV-H5Nx).

Characteristics	Category	Number (%) of IAV-H5Nx-Seropositive Cats in Category	*p*-Value	Odds Ratio (95% Confidence Interval)
Sex	Females Males	6/425 (1.4) 17/407 (4.2)	0.015	Reference category 3.04 (1.19, 7.80)
Castration (N = 188)	Intact Castrated	0/53 (0) 7/135 (5.7)	0.194	Reference category 6.25 (0.35, 111)
Breed	Pedigree DSH/DLH ^a^	2/168 (1.2) 21/664 (3.2)	0.164	Reference category 2.71 (0.63, 11.7)
Age category	>8 years ≤8 years	4/416 (1.0) 19/416 (4.6)	0.002	Reference category 4.93 (1.66, 14.6)

^a^ Domestic shorthair/domestic longhair cats.

**Table 4 viruses-17-00855-t004:** Multivariable logistic regression analysis of association between individual characteristics of cats and seropositive status for infection with influenza A virus H5Nx subtype (IAV-H5Nx).

Variables	Regression Coefficient (Standard Error)	Wald’s Statistic	*p*-Value	Odds Ratio (95% Confidence Interval)
Intercept	−5.28 (0.61)	-	-	-
Male sex	1.08 (0.48)	4.99	0.026	2.94 (1.14, 7.56)
Age ≤ 8 years	1.57 (0.56)	7.96	0.005	4.80 (1.61, 14.3)

Hosmer–Lemeshow χ^2^ test: χ^2^-statistic = 0.02, *p* = 0.990; Nagelkerke’s pseudo-R^2^ coefficient = 0.088.

## Data Availability

The raw data supporting the conclusions of this article will be made available by the authors on request.

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
