# Peer review of "The Seroprevalence of Influenza A Virus Infections in Polish Cats During a Feline H5N1 Influenza Outbreak in 2023"

_viruses, 2025, doi:10.3390/v17060855_

Round 1
Reviewer 1 Report
Comments and Suggestions for Authors
A manuscript (viruses-3591022) entitled "Seroprevalence of Influenza A Virus Infections in Polish Cats during a Feline H5N1 Influenza Outbreak in 2023" by Anna Golke et al. describes the seroprevalence of 835 cats to influenza A viruses and a subtype H5 influenza A virus using the commercially available ELISA kits. Authors indicated that sixty-eight of 835 cats were influenza A positive, and 23 cats among them were H5 influenza A positive. Though the authors did not provide information on influenza A virus shedding from cats nor on the cleavability of hemagglutinin (HA) protein to HA1 and HA2, this finding contributes to the risk estimation of H5 influenza virus emergence from avian to mammalian and the future emergence of human-to-human infection. The findings are worth to be considered for publication in Viruses.
However, this reviewer thinks that the authors need more clarification before publication. This reviewer raised several major and minor issues which the authors should address.
Major comments
- Materials subsection 2.3 is hard to understand in this study's context. Please clarify the way how three scenarios concern this study.
- Results subsection 3.1 should be moved to the Materials and Methods section because subsection 3.1 describes the cats from which blood was taken.
- The statement regarding the data handling needs more clarification. Doubtful cat sera (3/835) in NA antigen IAV-ELISA were added in the positive sera category and were counted as the apparent seroprevalence (line 207-209; 68+3=71). However, they (3/835) were not further tested for HA5 antigen. Only sixty-eight of 835 cat sera were tested for H5-ELISA (Table 3). A sentence (lines 217-219) should be clearer with Table 2.
- The discussion section is wordy compared to the obtained serological results. No direct evidence regarding the transmission of influenza from avian to cat, nor human to cat or vice versa, is provided from this study. H5 segment sequence of shedding virus from cats is required.
- This manuscript may have ethical issues. One is the informed consent from the cat owners to use blood for this serological study. There is no description in the "Informed Consent Statement". Second is the COI issue. One of the coauthors, Stepanie Lesceu, belongs to Innovative Diagnostics. The authors declared that IDvet Innovative Diagnostics financially supported ELISA tests in "Funding." Also, the authors stated that the funders had no role in the study design in a "Conflict of Interest." However, Stepanie Lesceu is shown to contribute to the conceptualization of the study in "Author Contributions." The authors should clear these inconsistent issues.
Minor comments
- Lane 30: It is described that 837 serum samples from domestic cats, but others showed 835 cats (lines 195, 197, 200, 209, 218, and 219).
- Lanes 49 and 51: The term "pandemic" is not proper because it does not occur globally. Please consider using "outbreak" or "epizootic".
- Lane 46: A term "influenza A viruses" is defined as (IAVs) here. "Influenza A virus" is still used instead of "IAV" (lanes 99, 110, and 326). Please unite the term.
- Lanes 46-47: The term "highly pathogenic avian influenza (HPAI) H5N1" is defined here. Later, two terms are used. One is "HPAI A (H5N1)" (lines 72 and 103). The second is "HPAI H5N1" (lines 70, 75, 79, and 125). Please unite the term.
- Lane 52: The term "genomic" is improper because a reassortment of influenza virus occurs between the same segments. Please consider deleting it.
- Lanes 78-79: It is described that the disease (HPAI H5 infection) in cats was usually severe and rapidly progressing, ranging from 1 - 7 days from the onset of clinical signs to death. If all infected cats died, serum obtained from the healthy cats makes no sense for studying the seroprevalence for HPAI H5. Please consider modifying the sentence.
- Lane 132: Special reagents and instruments used in the study should be shown in parenthesis with the production/selling company, city, country, or state if it is US. “Vetlab sp. Z o.o.” should be shown with city and country like (Vetlab sp. Z o.o. Wrochaw, Poland).
- Lanes 136-139: The meaning of sentences is not clear. Please describe more clearly.
- Lanes 141-142: A sentence seems not to be needed here. Please consider deleting a sentence.
- Lane 152: BioTek was changed to Agilent. Please describe like (Epoch Microplate Spectrophotometer, Agilent, Santa Clara, CA).
- Lane 204: Please check if the reference of Table 1 is correct or not.
- Lanes 227-228, Table 2: Please consider adding the age information of cats as shown in the bottom column of Table 3.
- Lanes 242-243, Table 3: Please check if the female and male numbers of 425 and 407 are correct or not. If they are correct, please explain why they are not 427 and 408, as in the second "Sex" column of Table 2.
- Lanes 242-243, Table 3: Please check if the DSH/DLH number of 664 in the Breed column is correct or not. If it is correct, please explain why they are not 667, as in the bottom column of Table 2.
- Lane 233, Figure 2: It is described that the median age of H5-seronegative cats was 9 years. However, the median value in IAV-seronegative cats in Figure 2 does not indicate 9. Please check the median plot in Figure 2.
- Lanes 242-243: Please explain the reason why the denominator of the Age category column is not 835 (total cat sera number of this study) but 416.
- Lane 248-250, Figure 1: Please indicate the meaning of the closed circle shown in every plot. Does it show a geometric mean value or median value?
- Lane 248-250, Figure 1: Please explain the reason why the total number of "n" is not 835 (total cat sera number of this study) but is 832 (211+205+196+220=832).
- Lane 251-252, Figure 2: Please explain the reason why the total number of "n" is not 835 (total cat sera number of this study) but is 832 (23+45+764=832).
- Lane 320: It is described that the anti-H5 seroprevalence in Polish domestic cats was relatively high (23/835=028). Please explain the grounds for considering "high."
Reviewer 2 Report
Comments and Suggestions for Authors
The manuscript relies primarily on a competitive ELISA to detect NP antibodies, followed by an H5-specific assay. While the study is scientifically sound and addresses a highly relevant topic, it requires minor to moderate revisions to strengthen its methodological transparency, refine speculative interpretations, improve the clarity of the Discussion, and ensure the tables and figures are fully self-explanatory.
ELISA is an appropriate initial screening tool; however, it is insufficient on its own if the objective is to gain detailed epidemiological insights or confirm subtype-specific exposure. Although the Discussion is generally comprehensive, well-contextualized, and highlights key public health and veterinary concerns regarding IAV infections in domestic cats, the authors should more clearly acknowledge the methodological limitations inherent in their serological approach.
Some sentences, particularly in the Discussion (e.g., lines 311–315), are overly long and could benefit from being divided into shorter, more digestible segments. The manuscript also includes plausible but unverified hypotheses, for example, regarding behavioral differences as risk factors, that should be framed more cautiously. The authors are encouraged to propose targeted future research to investigate these aspects more rigorously.
Additionally, while tables and figures are referenced appropriately in the text, their legends lack sufficient detail. Each table and figure should be comprehensible as a standalone element, without requiring the reader to refer to the main text. Legends should define all abbreviations, describe sample sizes and groups, and explain any symbols or statistical indicators used.
In summary, the manuscript is a valuable contribution to the field, but revisions are necessary to enhance its clarity, transparency, and usability for the reader.
Round 2
Reviewer 1 Report
Comments and Suggestions for Authors
A manuscript (viruses-3591022) entitled “Seroprevalence of Influenza A Virus Infections in Polish Cats during a Feline H5N1 Influenza Outbreak in 2023” by Anna Golke et al. is revised carefully according to the suggestions and the comments by the reviewer. The authors’ efforts made the study story more transparent and understandable than the initial version. However, this reviewer still has minor concerns that need to be clarified by the authors before accepting for publication in the journal “Viruses”.
Minor comments
- Lines 80-81: This reviewer asked to modify a sentence, but the author left it as before. It is described that the disease (HPAI H5 infection) in cats was usually severe and rapidly progressing, ranging from 1 - 7 days from the onset of clinical signs to death. If all infected cats died, serum obtained from the healthy cats would make no sense for studying the seroprevalence of HPAI H5. Please consider changing the sentences, for example, like the following;
- (line 63) this virus often induces “asymptomatic” infection
- (line 80) The “symptomatic infection” in cats was usually severe and rapidly progressing
- (line 90) resulting in the death of 38 out of 40 “symptomatic” cats
- (line 131) “asymptomatic”/mild/unrecognized infections
- Line 107: This reviewer asked to unify the defined term. The term “highly pathogenic avian influenza (HPAI) H5N1”is defined in line 49. Please use HPAI H5N “1”.
- Line 149: Please add the sentence indicating the use of the cat serum bank obtained during the routine diagnostic procedure, like in lines 416-417.
- Line 163: Please delete either IAV from the “IAV nucleoprotein of IAV”.
- Lines 184 and 287: Please change “seroprevalence” to “seropositivity”.
- Lines 292-293, Figure 2: It is described that the median age of H5-seronegative cats was 9 years. However, the median value in IAV-seronegative cats in Figure 2 does not indicate 9 years, while others correctly indicate 8 and 12 years. Please consider describing the median age as 8.0, 8.x, or 12.0 years; otherwise, draw the line on 9 exactly.
- Lines 416-417: If possible, please indicate the control number of the institutional agreement for using the cat serum bank obtained during the routine diagnostic procedure.
